# Joint Effect of Heavy Vehicles and Diminished Light Conditions on Paediatric Pedestrian Injuries in Backover Crashes: A UK Population-Based Study

**DOI:** 10.3390/ijerph191811689

**Published:** 2022-09-16

**Authors:** Bayu Satria Wiratama, Li-Min Hsu, Yung-Sung Yeh, Chia-Che Chen, Wafaa Saleh, Yen-Hsiu Liu, Chih-Wei Pai

**Affiliations:** 1Graduate Institute of Injury Prevention and Control, College of Public Health, Taipei Medical University, Taipei 110, Taiwan; 2Department of Biostatistics, Epidemiology, and Population Health, Faculty of Medicine, Public Health and Nursing, Universitas Gadjah Mada, Yogyakarta City 55281, Indonesia; 3Department of Surgery and Traumatology, National Taiwan University Hospital, Taipei 100, Taiwan; 4Department of Emergency Medicine, Faculty of Post-Baccalaureate Medicine, College of Medicine, Kaohsiung Medical University, Kaohsiung 807, Taiwan; 5Division of Trauma and Surgical Critical Care, Department of Surgery, Kaohsiung Medical University Hospital, Kaohsiung Medical University, Kaohsiung 807, Taiwan; 6Division of Colorectal Surgery, Department of Surgery, Taipei Medical University Hospital, Taipei Medical University, Taipei 110, Taiwan; 7Transport Research Institute, Edinburgh Napier University, Scotland EH11 4DY, UK

**Keywords:** paediatric pedestrian injuries, backover crashes, heavy vehicle, paediatric’s road safety, diminished light condition, logistic regression models

## Abstract

Backover crashes cause considerable injuries especially among young children. Prior research on backover crashes has not assessed the joint effect of heavy vehicles and diminished light conditions on injuries. By analysing the United Kingdom STATS19 crash dataset from 1991 to 2020, this study focused on backover crashes involving paediatric cyclists or pedestrians aged ≤17 years and other motorised vehicles. By estimating the adjusted odds ratio (AOR) of multiple logistic regression models, pedestrians appeared to have 82.3% (95% CI: 1.78–1.85) higher risks of sustaining killed or serious injuries (KSIs) than cyclists. In addition, casualties involved in backover crashes with heavy vehicles were 39.3% (95% CI: 1.35–1.42) more likely to sustain KSIs than those involved in crashes with personal cars. The joint effect of heavy vehicles and diminished light conditions was associated with a 71% increased probability of sustaining KSIs (AOR = 1.71; 95% CI: 1.60–1.83). Other significant joint effects included young children (aged 0 to 5 years) as pedestrian (AOR = 1.92; 95% CI: 1.87–1.97), in diminished light conditions (AOR = 1.23; 95% CI: 1.15–1.31), and with heavy vehicle (AOR = 1.37; 95% CI: 1.28–1.47).

## 1. Introduction

According to the World Health Organization, more than half of all road traffic fatalities involve vulnerable road users, such as pedestrians, cyclists, and motorcyclists [1]. Approximately 270,000 pedestrian deaths occur each year, accounting for 22% of 1.25 million road traffic fatalities. In some countries, this proportion is as high as two-thirds of all road traffic deaths. In the United States, 6205 pedestrians died and a pedestrian was killed every 85 min in traffic crashes in 2019 [2].

In the United States, approximately 227,000 backover crashes were reported to the police in 2014 [3]. The US federal government estimated that in backover crashes, approximately 15,000 people are injured each year, of whom 210 people die, with many of them aged less than 5 years [3,4,5]. Globally, an estimated 80% of paediatric pedestrian deaths in backover crashes occur at nonintersections, with a large proportion involving children under the age of 5 years [6,7].

Although backover crashes account for a small part of pedestrian–motor vehicle crashes, such crashes cause considerable injuries, especially among young children [8,9,10]. A backover crash typically occurs at low speeds, in which a vehicle coming out of a driveway or parking lot backs over an unattended pedestrian. A Canadian study revealed that children were often exposed to the risk of backover crashes in areas outside the lane [9], and the resultant injuries were severe [9,10,11,12].

To reduce backover crashes and the resulting injuries, relevant preventive interventions must target pedestrians of different ages and locations [9]. Technologies adopted to prevent backover crashes include rear-view cameras and rear parking sensors. Over the past years, studies have examined the effectiveness of rear-view cameras and rear parking sensors in reducing backover crashes in the United States and Japan [12,13]. The recent emergence of automatic reverse braking has resulted in the reduction in backover crash rates by 78% [14,15]. In the United States, a rear-view camera system that complies with specific rear vision specifications has become mandatory for new light passenger vehicles from May 2018.

A literature review suggests that backover crashes have a substantial impact on pedestrian safety, especially paediatric pedestrian safety. Heavy vehicles, with larger blind spots than private cars and which are not mandated to have rear-view cameras or sensors, are particularly hazardous to paediatric road users. Pedestrians, who are exposed to a reversing vehicle, may be at higher risks of injuries under diminished light conditions due to decreased visibility. Thus far, prior studies on backover crashes have not assessed the joint effect of heavy vehicles and diminished light. This study analysed the UK STATS19 crash dataset to examine the joint and individual associations of heavy vehicles and diminished light conditions with paediatric injury in backover crashes.

## 2. Materials and Methods

### 2.1. Data

This research examined the UK STATS19 crash dataset from 1991 to 2020. STATS19 contains data on every crash resulting in personal injury that is reported to the UK police within 30 days. Since 1979, data on crashes, vehicles, and victim characteristics have been annually collected. This study was approved by the Taipei Medical University Joint Institutional Review Board (N202011030).

### 2.2. Casualties

This study focused on backover crashes involving cyclists or pedestrians aged ≤17 years and motorised vehicles (excluding motorcycles). Figure 1 presents the flowchart of data selection. We omitted victims with missing data on sex, age, speed limit, crash time, weather condition, light condition, or crash partner vehicle type. A complete case analysis approach, which was proposed by Kang [16], was used in this study. No difference between cases with and without missing data was observed (*p* > 0.05). A total of 467,465 paediatric casualties were included in the final dataset.

### 2.3. Outcome and Variable Definitions

This study defined injuries as either killed or seriously injured (KSI) or mild injuries. Mild injuries included sprains (including whiplash), bruises, cuts, and mild shock requiring roadside assistance. Injuries not requiring medical treatment were also classified as mild injuries. Severe injuries were those resulting in hospitalization or were any of the following injuries: fracture, concussion, internal injuries, crushing, and severe shock requiring medical treatment. Fatal injuries were those resulting in death within 30 days of the accident. Fatal and severe injuries together constituted the standard metric of killed or seriously injured (KSI). This study collected data on crashes, casualty characteristics, and crash partner characteristics. Casualty and crash partner characteristics were sex (male or female casualty or crash partner) and age (casualty: 0–5, 6–10, 11–14, and 15–17 years; crash partner <18, 18–40, 41–64, and ≥65 years). Casualty type was defined as a cyclist or a pedestrian, and crash partner vehicle types were personal car, taxi, heavy vehicle, or other types. Crash time was classified as either nighttime (sunset to sunrise) or daytime (sunrise to sunset). The weather was categorized as either fine or adverse. The crash location was determined from speed limit data; crash locations (roads) with speed limits of <30 miles per hour were considered urban, and those with speed limits of ≥30 miles per hour were considered rural [17]. In the UK STATS19 database, light condition was classified into five categories: daylight; darkness: street light present and lit; darkness: street light present and unlit; darkness: no street lighting; darkness: street lighting unknown. We further merged these five categories to an optimal (daylight) or diminished light condition (other categories than daylight). This study also collected and classified data on the day type (weekdays, weekends, and public holidays). Based on a previous study, 10 types of public holidays are recognized in the UK: bank holidays, New Year’s Day, Good Friday, Easter Monday, summer bank holidays, spring bank holidays, Christmas, Boxing Day, the Queen’s 2002 Golden Jubilee, and the Queen’s 2012 Diamond Jubilee. The exact dates for each public holiday vary across each year. The specific dates for each holiday can be found on https://www.gov.uk/bank-holidays (accessed on 10 April 2022).

### 2.4. Statistical Analysis

First, we compared the distributions of casualties across age, sex, lighting condition, weather condition, road type, vehicle type, day type, crash time, and injury severity (KSI or mild injury). As in a previous study [17,18,19], we utilized a *p* value of <0.2 as the cut-off point for including independent variables in a multiple logistic regression model. Multiple logistic regression was used to estimate adjusted odds ratios (AORs) between potential risk factors for KSIs. Cramér’s *V* was used to assess multicollinearity between independent variables. An alpha of 0.05 was used, yielding a confidence level of 95%. Missing data were considered to be at least missing at random and were, therefore, excluded from the analysis [16]. This study analysed the joint effect of the following variables of interest: casualty’s age (0–5 vs. 6–17 years), casualty type (pedestrians vs. cyclists), day type (public holidays or weekends vs. weekdays), crash time (nighttime vs. daytime), light condition (diminished vs. optimal), and crash partner vehicle type (heavy vehicle vs. other types). This study used Strengthening the Reporting of Observational Studies in Epidemiology guidelines for reporting the joint effects [20].

## 3. Results

Table 1 presents the casualty characteristics. A total of 467,465 casualties were included; of them, 73.11% involved pedestrians, and 26.89% involved cyclists. As many as 99,008 (21.18%) and 368,457 (78.82%) casualties sustained KSIs (21.18%) and mild injuries (78.82%), respectively. The backover crashes occurring during daytime (79.53%) outnumbered those occurring at nighttime (20.47%). KSIs were more prevalent among casualties aged 0–5 years (23.55%) than those among other age groups. Pedestrians had a higher proportion of KSIs (23.56%) than cyclists (14.70%). The analysis of crash partner vehicle type revealed the highest proportion of KSIs in casualties involved in backover crashes with heavy vehicles (26.72%) compared with those involved in crashes with personal cars (20.67%), taxis (21.60%), and other vehicle types (25.19%). The proportion of KSIs among casualties involved in backover crashes under diminished lighting conditions was higher (25.85%) than those involved in crashes in optimal lighting conditions (20.30%).

Table 2 lists the results of the multiple logistic regression model for KSIs among casualties involved in backover crashes. Casualties aged 0–5 years had the highest risks of sustaining KSIs (AOR = 1.074; 95% CI = 1.046–1.102) over those in other age groups. Pedestrians had 1.823 (AOR = 1.823; 95% CI = 1.789–1.857) times higher risks of sustaining KSIs than cyclists. Male casualties were 12.6% (AOR = 1.126; 95% CI = 1.109–1.144) more likely to sustain KSIs than female casualties. Casualties involved in backover crashes with heavy vehicles had an increased probability of sustaining KSIs by 39.3% (AOR = 1.393; 95% CI = 1.357–1.429) compared with those involved in backover crashes with personal cars. Other risk factors for KSIs were a diminished light condition (AOR = 1.237; 95% CI = 1.210–1.265), fine weather (AOR = 1.157; 95% CI = 1.127–1.188), public holiday (AOR = 1.107; 95% CI = 1.052–1.165), weekend (AOR = 1.125; 95% CI = 1.106–1.145), and nighttime (AOR = 1.167; 95% CI = 1.142–1.193).

The results of the analysis of the joint effect of casualty age, casualty type, day type, crash time, light condition, and crash partner vehicle type are illustrated in Figure 2. The joint effect between the crash partner vehicle type and light condition was a statistically significant factor. Casualties involved in backover crashes with heavy vehicles and diminished light conditions had 71% (AOR = 1.71; 95% CI = 1.60–1.83) higher risks of sustaining KSIs than those involved in crashes with other vehicle types and optimal light conditions. Furthermore, pedestrians involved in backover crashes with heavy vehicles were 133% (AOR = 2.33; 95% CI = 2.26–2.41) more likely to sustain KSIs than cyclists involved in backover crashes with other vehicle types. Young children aged 0 to 5 years had higher risks of sustaining KSIs as pedestrians (AOR = 1.92; 95% CI = 1.87–1.97) and in backover crashes occurring on public holidays or during the weekend (AOR = 1.19; 95% CI = 1.15–1.23), at nighttime (AOR = 1.16; 95% CI = 1.10–1.23), under diminished light conditions (AOR = 1.23; 95% CI = 1.15–1.31), and with heavy vehicles (AOR = 1.37; 95% CI = 1.28–1.47).

## 4. Discussion

We found that the proportion of KSIs among children injured in backover crashes was 21.18%, implying that approximately 21 of 100 children sustained KSIs in backover crashes. This number is higher than those reported in previous studies (e.g., Fenton et al. [11]) but lower than those reported by Zonfrillo et al. [6] and Pikney et al. [21]. Furthermore, paediatric pedestrians were more severely injured than paediatric cyclists. This may be because paediatric pedestrians are less conspicuous than paediatric cyclists, thereby indicating the requirement of improving the visibility of paediatric pedestrians [22,23].

Our research indicated that children aged 0–5 years had higher risks of sustaining KSIs than those aged 6–17 years. Previous studies have similarly concluded that young children had the highest risks of sustaining fatal injuries in backover crashes compared with those in other age groups [11,21,24,25,26,27,28]. This may be attributed to the inability of young children to recognize environmental hazards [11,29], the limited visibility of children to drivers [11,29], higher mean injury severity score [11,28], longer intensive care unit stay [11,28], and the increased probability of head injuries [11].

The joint effect analysis revealed that young children had high risks of sustaining KSIs when involved in backover crashes with heavy vehicles, as pedestrians, and under diminished light conditions. Potential interventions should primarily target parents of children aged 0 to 5 years and drivers of heavy vehicles. Moreover, the mandatory equipment of heavy vehicles with reversing sensors may constitute an effective countermeasure.

The risk of sustaining KSIs was higher at nighttime than at daytime; this finding corroborates with those of previous studies [11,17]. This indicates the detrimental effect of diminished light conditions on KSIs among children. The increased risk of crashes at nighttime is due to impaired motion perception at night [17]. Furthermore, the driver’s visibility is limited at night, which further increases the risk of KSIs among children involved in backover crashes [11,29].

Our research contributes to the literature on paediatric road injuries by identifying the joint effect of heavy vehicles and diminished light conditions. Children involved in backover crashes with heavy vehicles and under diminished light conditions exhibited higher risks of sustaining KSIs than those involved in crashes with other vehicles and in optimal light conditions. This may be because heavy vehicles are generally not equipped with reversing sensors and have wider blind spots than other vehicle types [7,21,26,30]. Therefore, diminished light conditions may limit drivers’ rear visibility, thereby increasing the injuries among casualties.

Our finding corroborates with previous research that fine weather was associated with higher risks of KSIs than adverse weather [17,18]. This finding may be due to the fact that visibility and traffic/driving condition can be better during fine weather than those during adverse weather. Drivers may as a result compensate the increased visibility with an increased speed and reduced concentration [31].

## 5. Conclusions

We identified the following statistically significant joint effects that could explain paediatric injuries in backover crashes: heavy vehicles and diminished light condition; young child age and diminished light condition; and young child age and heavy vehicles. Interventions should focus on mandating reversing sensors for heavy vehicles. This research has several limitations. First, data on vehicles with or without reversing sensors were not available. Therefore, the association between heavy vehicles and the high risk of KSIs should be explored further by using data from reversing sensors. We assumed that compared with private vehicles, fewer heavy vehicles are equipped with reversing sensors. Second, in general, backover crashes occur in private parking lots or driveways [7], and such crashes may be underreported in the police database. The use of databases, such as hospital or insurance databases, together with clinical data may provide an accurate estimation of injuries sustained by those involved in backover crashes. Finally, this study could not control for additional variables, such as specific crash location, vehicle speed, smartphone use, visibility, ambulance response time, vehicle volume data, and distance to the trauma centre, that may considerably affect injury severity among casualties involved in backover crashes.

## Figures and Tables

**Figure 1 ijerph-19-11689-f001:**
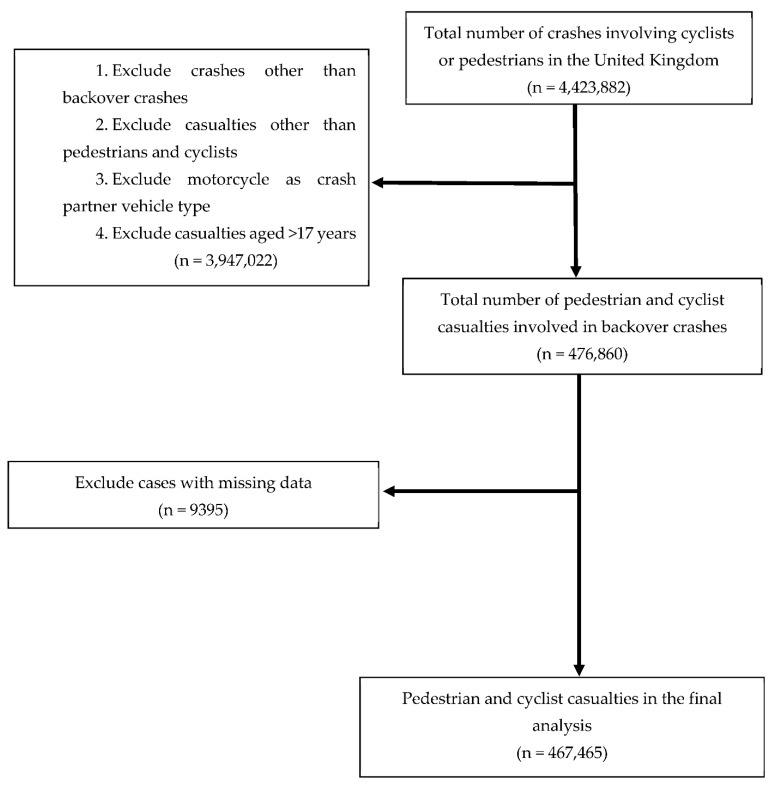
Sample selection process.

**Figure 2 ijerph-19-11689-f002:**
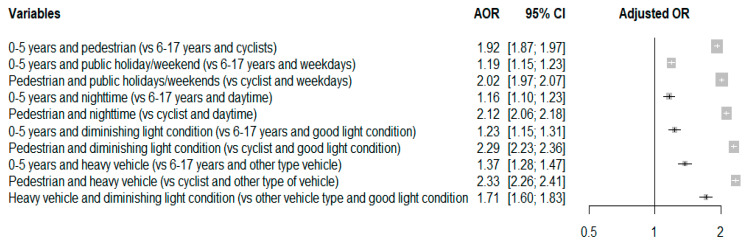
Joint effects of variables.

**Table 1 ijerph-19-11689-t001:** Distribution of study participant’s characteristics based on injury status.

Variables	Pedestrian and Cyclist Injuries			
	KSIs	Non-KSIs	Total	*p* Value
	n (%)	n (%)	n (%)	
Casualty’s age				
15–17 years	18,148 (20.66)	69,683 (79.34)	87,831 (18.79)	<0.001
11–14 years	36,514 (21.06)	136,846 (78.94)	173,360 (37.09)	
6–10 years	29,819 (20.62)	114,765 (79.38)	144,584 (30.93)	
0–5 years	14,527 (23.55)	47,163 (76.45)	61,690 (13.20)	
Driver’s age				
≥65 years	99,008 (20.14)	23,652 (79.86)	29,616 (6.34)	<0.001
41–64 years	33,071 (20.02)	132,142 (79.98)	165,213 (35.34)	
18–40 years	58,438 (21.91)	208,263 (78.09)	266,701 (57.05)	
<18 years	1535 (25.86)	4400 (74.14)	5935 (1.27)	
Casualty’s sex				
Female	33,016 (21.14)	123,151 (78.86)	156,167 (33.41)	0.65
Male	65,992 (21.20)	243,306 (78.80)	311,298 (66.59)	
Driver’s sex				
Female	31,396 (19.50)	129,618 (80.50)	161,014 (34.44)	<0.001
Male	67,612 (22.06)	238,839 (77.94)	306,451 (65.56)	
Casualty type				
Pedestrian	80,526 (23.56)	261,246 (76.44)	341,772 (73.11)	<0.001
Cyclist	18,842 (14.70)	107,211 (85.30)	125,693 (26.89)	
Crash partner vehicle type				
Personal car	87,121 (20.67)	334,265 (79.33)	421,386 (90.14)	<0.001
Taxi	1546 (21.60)	5610 (78.40)	7156 (1.53)	
Heavy vehicle	9366 (26.72)	25,687 (73.28)	35,051 (7.50)	
Other vehicle	975 (25.19)	2895 (74.81)	3870 (0.83)	
Crash location				
Rural (≥30 mile/h)	97,316 (21.20)	361,807 (78.80)	459,123 (98.22)	0.043
Urban (<30 mile/h)	1692 (20.28)	6650 (79.72)	8342 (1.78)	
Weather condition				
Adverse weather	12,552 (20.15)	49,752 (79.85)	62,304 (13.33)	<0.001
Fine weather	86,456 (21.34)	318,705 (78.66)	405,161 (86.67)	
Light condition				
Diminished	19,178 (25.85)	55,007 (74.15)	74,185 (15.87)	<0.001
Optimal	79,830 (20.30)	313,450 (79.70)	393,280 (84.13)	
Road condition				
Bad	21,773 (21.56)	79,205 (78.44)	100,978 (21.60)	0.001
Good	77,235 (21.07)	289,252 (78.93)	366,487 (78.40)	
Day type				
Public holiday	2004 (22.28)	6989 (77.72)	8993 (1.92)	<0.001
Weekend	22,881 (22.46)	79,015 (77.54)	101,896 (21.80)	
Weekday	74,123 (20.79)	282,453 (79.21)	356,576 (76.28)	
Crash time				
Nighttime	23,740 (24.81)	71,946 (75.19)	95,686 (20.47)	<0.001
Daytime	75,268 (20.25)	296,511 (79.75)	371,779 (79.53)	

**Table 2 ijerph-19-11689-t002:** Multivariate analysis using multiple logistic regression.

Variables	Adjusted Odds Ratio (AOR)	*p* Value	95% CI
Casualty’s age			
0–5 years	1.074	<0.001	1.046–1.102
6–10 years	0.996	0.756	0.975–1.018
11–14 years	1.058	<0.001	1.037–1.081
15–17 years	Reference		
Driver’s age			
≥65 years	0.937	<0.001	0.908–0.965
41–64 years	0.901	<0.001	0.887–0.915
<18 years	1.190	<0.001	1.122–1.263
18–40 years	Reference		
Casualty’s sex			
Male	1.126	<0.001	1.109–1.144
Female	Reference		
Driver’s sex			
Male	1.098	<0.001	1.082–1.116
Female	Reference		
Casualty type			
Pedestrian	1.823	<0.001	1.789–1.857
Cyclist	Reference		
Crash partner vehicle type			
Heavy vehicle	1.393	<0.001	1.357–1.429
Taxi	0.871	0.311	0.916–1.028
Other vehicle	1.243	<0.001	1.144–1.338
Personal car	Reference		
Crash location			
Rural (≥30 mile/h)	1.054	0.057	0.998–1.113
Urban (<30 mile/h)	Reference		
Weather condition			
Fine weather	1.157	<0.001	1.127–1.188
Adverse weather	Reference		
Light condition			
Diminished light condition	1.262	<0.001	1.235–1.289
Optimal light condition	Reference		
Road condition			
Bad road condition	1.022	0.049	0.999–1.044
Good road condition	Reference		
Day type			
Public holiday	1.105	<0.001	1.050–1.162
Weekend	1.122	<0.001	1.103–1.141
Weekday	Reference		
Crash time			
Nighttime	1.167	<0.001	1.142–1.193
Daytime	Reference		

## Data Availability

The current research used the STATS19 database, which contains data on all road traffic accidents in the United Kingdom.

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
