# Peer review of "Joint Effect of Heavy Vehicles and Diminished Light Conditions on Paediatric Pedestrian Injuries in Backover Crashes: A UK Population-Based Study"

_ijerph, 2022, doi:10.3390/ijerph191811689_

Round 1
Reviewer 1 Report
This study focuses on the effects of backover crashes; more specifically, attention is on crashes involving pediatric cyclists and pedestrians. A multiple logistic regression model is carried out. The authors can find below a list of comments and suggestions.
I suggest uniforming the numbers of decimal digits of CI in the abstract.
I suggest including in the Keywords more widespread words that are common to the general subject discipline and could help the diffusion of the article (such as “road safety”, “pedestrians safety”…) and at least one keyword related to the method used in the study (such as “logistic regression models”);
On line 42 it is reported that “6,205 pedestrians died and a pedestrian was killed every 85 min in traffic crashes in 2019”; the statement is provided with the reference [2] which is the 2018 Global Status Report on Road Safety…Please correct either the statement in the manuscript or the reference.
On line 128 you write as follows: “The light condition was classified as an optimal or diminished light condition.” How you did you obtain this information? Was light condition indicated in the crash database in two ways, optimal and diminished?
Are lighting conditions correlated only to backover crashes occurred during night-time?
I suggest to report also the date when you indicate the types of public holidays; this information could be useful for researchers who want to repeat your study (it is more international information).
I suggest modifying the title of Table 1: it seems this caption does not describe the table content.
Please, improve the quality of Figure 2. It is difficult to see and observe the graph.
I suggest moving consideration on the limitations of the study (lines 242-254) from Discussion section to the conclusion section.
Reviewer 2 Report
The paper describes an analysis of crash data regarding backover crashes. Given the results the authors give some recommendations for preventing this type of crashes.
Figure 1 on page 3
This is not really a flowchart: the arrows do not show the right directions for a flow.
Lines 153/154
I presume you do not have data about vehicle volumes. The difference between daytime and night time is presumably much higher if volume data could be taken into account.
Line 184
The finding regarding more crashes in ‘fine weather’ is not elaborated. Is this not contradictory to in particular the light condition?
Line 207
The percentage o f 21.18 mentioned in this section differs from the percentage in line 155 (23.55%).
Line 212
You recommend to improve the visibility of young pedestrians. Their dimensions (especially their height) cannot be changed. And a yellow coat will not really make much difference.
Line 246
A speculation does not belong in a paper.
Line 259/262
There is much evidence that education does not have effect on road safety. See: Dragutinovic, N. & Twisk, D. The effectiveness of road safety education. SWOV Institute for Road Safety, Leidschendam, The Netherlands.
